# UAV-Based Servicing of IoT Nodes: Assessment of Ecological Impact

**DOI:** 10.3390/s23042291

**Published:** 2023-02-18

**Authors:** Jarne Van Mulders, Jona Cappelle, Sarah Goossens, Lieven De Strycker, Liesbet Van der Perre

**Affiliations:** ESAT-DRAMCO, Ghent Technology Campus, KU Leuven, 9000 Ghent, Belgium

**Keywords:** Internet of Things, energy efficiency, wireless power transfer, sustainability

## Abstract

Internet of Things (IoT) nodes get deployed for a variety of applications and often need to operate on batteries. This restricts their autonomy and/or can have a major ecological impact. The core idea of this paper is to use a unmanned aerial vehicle (UAV) to provide energy to IoT nodes, and hence prolong their autonomy. In particular, the objective is to perform a comparison of the total energy consumption resulting from UAV-based recharging or battery replacement versus full provisioning at install time or remote RF-based wireless power transfer. To that end, an energy consumption model for a small license-free UAV is derived, and expressions for system efficiencies are formulated. An exploration of design and deployment parameters is performed. Our assessment shows that UAV-based servicing of IoT nodes is by far more beneficial in terms of energy efficiency when nodes at distances further than a few meters are serviced, with the gap increasing to orders of magnitude with the distance. Our numerical results also show that battery swapping from an energy perspective outperforms recharging in the field, as the latter increases hovering time and the energy consumption related to that considerably. The ecological aspects of the proposed methods are further evaluated, e.g., considering toxic materials and e-waste.

## 1. Introduction

The introduction of the Internet of Things (IoT) has created the opportunity to develop a myriad of applications. The number of IoT edge devices that will be deployed is predicted to grow fast, from 5–10 billion in 2020 up to 200+ billion in 2030 [1]. However, installing and maintaining these countless devices can be cumbersome, if not impossible, if done manually. It is evident that, for an efficient deployment of these sky-high numbers, any dedicated installation or maintenance effort should be minimized. The ‘fire and forget’ credo, towards which recent IoT devices are aimed, does not properly consider the End of Life (EoL), especially since toxic materials (e.g., in batteries) can remain in nature and cause pollution. From an ecological point of view, it would be problematic if even a small percentage of these battery-powered devices were not recovered from the environment. Therefore, systems for automatic battery charging/replacement and commissioning are indispensable.

Solid technological advances in the IoT have focused on low-power communication and design [2]. This has led to the development of energy-efficient long-range wide-area networks (LoRaWANs), strict energy management, load switching to reduce standby/sleep currents, and the use of low-power Microcontroller Units (MCUs). The lifetime of battery-powered devices can be extended by applying low-power strategies such as ‘think before you talk’, going to sleep as much as possible, etc. [2]. Applying these strategies has its limits, especially for time-critical applications, which require more frequent measurements and communication. As a result, there is a need for human intervention to recharge or replace the battery, or even replace the complete IoT device.

Automation is required for an efficient maintenance of massively deployed IoT devices to ensure their longevity. Lessons learned from early IoT adopters [3] include the existence of a ‘skills gap’ between what is currently required to deploy and maintain IoT systems versus the abilities of maintenance people. Automation would eliminate the need for technical knowledge and save time and money. Moreover, an IoT network can be applied in a variety of environments that can be difficult to reach for humans, e.g., environmental monitoring of remote field sites. Locations such as these bring practical challenges, including physical installation and maintenance difficulties.

The challenges regarding the autonomy of IoT nodes and the overhead involved in manual interventions to recharge or replace batteries motivate the main idea and objective of this paper: to study the opportunities in utilizing unmanned vehicles (UVs) to increase the lifespan of IoT devices. We present three scenarios, for each of which the energy budget is analyzed. These include noncoupled wireless power transfer (WPT) technologies and UV-based solutions. We assess the energy budget for UVs that serve as mobile power banks and show that this approach can lead to a more sustainable integration. All these scenarios create, in theory, an infinite autonomy and, moreover, enable a reduction in manual labor for use cases that cannot rely on their own source of harvested green energy.

The remainder of this paper is structured as follows. Section 2 discusses the energy delivery scenarios and associated design and deployment assumptions. A model for the energy consumption of a UV has been adapted based on actual measurements. This model is then used in the quantitative study. The overall energy budget calculations for each energy delivery scenario are explained in Section 3. Section 4 compares the three solutions by calculating their respective performances. Sustainability gains are demonstrated in Section 5, and Section 6 concludes this manuscript.

## 2. System and IoT Node Architecture and Related Work

Higher autonomy of battery-powered embedded devices can be achieved by making better use of the available energy. However, adopting storage technologies with higher energy densities can also result in lifespan gains due to the availability of more energy. Lithium thionyl chloride (LTC) batteries with extremely high energy densities of around ≈ 1000 Wh/l are often selected.

If longevity proves insufficient, harvesting energy from environmental sources is often explored as an interesting solution. Solar, thermal, and mechanical sources with power densities up to $10^{5}$, $10^{3}$, and $10^{3}$ μW/m^3^, respectively [2], are often sufficient to cover the full energy consumption of the applications. Where these solutions cannot be implemented or when the autonomy is still too limited, the final option is a human intervention to recharge or replace the drained battery or the whole device. This is labor-intensive and leads to high maintenance costs and extra traffic on the road. To avoid intensive maintenance, this article evaluates novel techniques for extending the lifetime of IoT devices, comparing the following three options:1.Using uncoupled wireless power transfer approaches such as laser power transfer or radio frequency (RF) power transfer to recharge the IoT node’s batteries. These techniques feature wireless power over longer distances combined with lower efficiencies compared with the coupled WPT techniques [4].2.Engaging UVs deployed as a mobile power bank to charge batteries on-site.3.Employing UVs acting as a technician to replace batteries autonomously.

Figure 1 shows the scenarios discussed in this article. To assess and compare system performance and clarify the benefits of each approach, design and deployment assumptions need to be made. Different types of UVs are available, e.g., unmanned aerial vehicle (UAV), unmanned ground vehicles (UGVs), or autonomous underwater vehicles (AUVs). This work focuses on a UAV. The UAV is expected to fly to the IoT applications and deliver the energy. We compare the radio frequency power transfer (RFPT) and UAV approaches departing or delivering from the same initial location, as illustrated in Figure 1. The distance between IoT node and both UAV charge station and the RFPT transmitter is assumed to be equal to the line-of-sight (LoS) distance. This work focuses on the remote charging of a single IoT node.

### 2.1. Uncoupled Wireless Power Transfer

We present a concise study of the effectiveness of lifetime enhancements of IoT nodes through uncoupled WPT techniques to serve as a reference for the UAV-based servicing options. In uncoupled systems, no magnetic or electrical coupling is present between the transmit and receive sides. The wireless power delivery happens over distances where the receiver is located in the far field of the transmitter. Moreover, it is feasible to power multiple devices at the same time, in contrast to coupled technologies. Uncoupled approaches fully rely on EM waves captured by the receiver device. In the literature, two technologies are distinguished to deliver power in the far field: RFPT and laser power transfer (LPT). In this comparative study, RFPT technology is considered, as it currently is the most convenient of the two in terms of requirements for an unobstructed path between transmitter and receiver.

The achievable distance using RFPT is restricted to some meters, due to the high path loss and limited sensitivity of the harvester circuits. This is further elaborated in Section 3.1. In most cases, such as with radio frequency identification (RFID) systems, this technology is selected with the aim of powering very low-energy-consuming nodes and rarely for substantially charging devices. The energy storage will rather consist of a storage buffer capacitor instead of a battery.

The main objective of our current study is to evaluate efficiency as a function of distance. Necessary assumptions are made in order to obtain quantitative estimations. The power output of the RFPT, specified as effective isotropic radiated power (EIRP), is limited to comply with regulatory European Telecommunications Standards Institute (ETSI) constraints [5]. Low-complexity models are used in a single-input single-output (SISO) configuration with a single-tone waveform, and a single frequency is considered. The power amplifier (PA) of the RFPT is assumed to have a fixed efficiency.

Related work. Recently, a lot of research has been conducted regarding waveform design to increase RF/DC efficiency [6] and multiband harvesting solutions [7] to increase harvested power. Distributed beamforming to increase the received power and coverage is investigated, a.o., in the European project REINDEER [8]. By coherent operation of the individual devices, a power spot at a certain location could be created. This RadioWeaves approach has already been introduced in [9].

### 2.2. UV with Charging Facility

Quite recently, approaches to energize nodes based on a UAV capable of transferring energy to an IoT node have been introduced. As such, this study aims to provide a proper estimate of UAV energy requirements in relation to the practical parameters, design choices, and various assumptions.

On site, the UAV should refill the application storage buffer with several hundred to thousands of joules of energy. Two scenarios can be assumed. The UAV first lands, then starts transferring charge (1), or hovers while delivering energy simultaneously (2). Both wireless and cable solutions would be available in these scenarios. Figure 2 summarizes the four procedures for transferring energy.

Providing a landing spot at every node is often not practical, hence this option is not further examined in this research. Charging during hovering offers more flexibility because no landing spot is required. Moreover, a wireless power transmission solution is opted for because of less stringent alignment constraints that do exist when using contacts. Article [10] already summarized the wireless power transfer approaches that could be implemented on a UAV. The fine-grained position system to align the node and UAV is out of the scope of this discussion. The underlined procedure in Figure 2 is elaborated in Section 3.2.

We further take into account the constraint of a small <250 g UAV, as for these no registration or training is required according to the European Union Aviation Safety Agency (EASA) [11]. The impact of wind and gusts in the UAV flight are neglected, and flights are considered to be performed at sea level.

Related work. Tiurlikova et al. [12] propose a UAV equipped with an inductive WPT link to charge LoRaWAN-enabled devices. Alternatively, Tiurlikova et al. [13] present a multi-UAV energy harvesting network with multiple UAVs transmitting energy to several nodes through RF links.

This article primarily focuses on the feasibility of UAV-based systems, with an emphasis on energy and achievable efficiency. Control of the UAV and routing (navigation and maintenance scheduling) algorithms, such as the ones elaborated in [14,15], are considered out of the scope of this paper.

### 2.3. IoT Energy Storage Replacement

In this scenario, we propose the replacement of the IoT battery by engaging a UAV. Here, the UAV will rather get the function of a packet delivery system, as described by, e.g., Pugliese et al. in [16]. The main difference is the payload consisting of an energy storage buffer that is mechanically and electronically compatible with the IoT battery slot, instead of a packet. The constraint of a 250 g UAV still applies, thus limiting carried weight. Similar to the option described in the previous section, a distinction between landing and hovering can be made. We further assume the worst-case scenario in terms of energy consumption, i.e., the UAV hovers during the swapping process. So, the UAV flies to the IoT location autonomously and swaps a battery of similar weight. The extra UAV power consumption related to the additional weight is taken into account in the model parameters in Section 2.4. The analysis is elaborated in Section 3.3.

### 2.4. UAV Energy Consumption Model

Scenarios 2 and 3 from Figure 1 require a UAV to transfer the energy. To properly estimate the overall energy in Section 3, the energy consumption of the self-constructed UAV, depicted in Figure 3, is needed. To obtain representative estimates of the power consumption and enable analysis and optimized system design and operation, a mathematical model is derived. This model for the power consumption should at least incorporate the impact of the weight (including payload) and speed of the UAV.

Related work. Previous research has shown that UAVs are difficult to model because of the large number of parameters and variables involved. A comprehensive analysis by Zhang et al. [17] present multiple solutions to model the UAV power consumption. In their work, the authors define four categories of parameters that influence the power drawn from the UAV battery: UAV design, the environment (flight conditions), UAV dynamics, and delivery operation. Furthermore, Zhang et al. [17] describe five fundamental models for steady-level flight. Some energy consumption models, such as the one by Dorling et al. [18], ignore the impact of a UAV’s speed on its energy consumption. Other models, such as the work of Ferrandez et al. [19], apply to much heavier UAVs, making them less representative for the use-case considered in this manuscript. Regression based on field trials may also be used to estimate UAV power consumption; however, this eliminates the flexibility of later changing, e.g., the payload weight. The model of Kirschstein [20] is the most comprehensive model and considers component-based parameters of the UAV. The parameters in a UAV component model are obtained by decomposing several individual forces of the UAV and describing them by separated models. The extended model from Kirschstein [20], based on previous research described in [21], estimates multiple constants from a set of equations, provided further in this paper, by conducting three experiments. The latter model considers vertical speed and airspeed combined with the total weight (incl. payload). The survey of Zhang et al. [17] shows that avionics power constitutes a significant share of the total consumption, especially in smaller UAVs. Our proposed model for the tiny, lightweight (170 g) UAV of Figure 3 should hence take into account avionics power, which is essentially the current draw of the onboard components without active motors. A more comprehensive analysis of UAV models is beyond the scope of this paper.

We hence based the UAV power consumption analysis on the model of Liu et al. [21], supplemented by the missing avionics power. This study assumes the UAV will be in steady-state flight most of the time, yielding zero net acceleration. The flight path of the UAV generally consists of a steady-state ascent, followed by a forward flight, and finally a descent. The total power $P_{\mathrm{uav}}$ drawn from the battery is decomposed into 4 components (Equation (Equation 1)) and is impacted mainly by the total mass *m*, the vertical and horizontal speed $V_{\mathrm{ver}}$ and $V_{\mathrm{hor}}$, and the thrust *T*, representing the amount of upward force that the UAV is able to generate: the induced power $P_{i}$ (Equation (Equation 2)), profile power $P_{p}$ (Equation (Equation 3)), parasitic power $P_{\mathrm{par}}$ (Equation (Equation 4)), and avionic power $P_{\mathrm{avio}}$. The induced power provides thrust by pushing air downwards, the profile power overcomes the rotational drag encountered by the rotating propeller blades, and the parasitic power accounts for the air drag generated in forward motion. In this model, the impact of wind is neglected, so that the airspeed equals the ground speed, denoted as $V_{\mathrm{hor}}$. The total trust *T* is represented by Equation (Equation 5), with α the angle of attack, *m* the total mass, and *g* the gravitational acceleration.
(1)$$\begin{array}{c} P_{\mathrm{uav}}=P_{\mathrm{avio}}+P_{i}+P_{p}+P_{\mathrm{par}} \end{array}$$
(2)$$\begin{array}{c} P_{i}\left(T,V_{\mathrm{vert}}\right)=k_{1}T\left[\frac{V_{\mathrm{vert}}}{2}+\sqrt{{\left(\frac{V_{\mathrm{vert}}}{2}\right)}^{2}+\frac{T}{k_{2}^{2}}}\right] \end{array}$$
(3)$$\begin{array}{c} P_{p}\left(T,V_{\mathrm{hor}}\right)=c_{2}T^{3/2} \end{array}$$
(4)$$\begin{array}{c} P_{\mathrm{par}}\left(V_{\mathrm{hor}}\right)=c_{3}V_{\mathrm{hor}}^{3} \end{array}$$
(5)$$\begin{array}{c} T=\sqrt{{\left(mg-c_{4}{\left(V_{\mathrm{hor}}\cos\alpha \right)}^{2}\right)}^{2}+{\left(c_{3}V_{\mathrm{hor}}^{2}\right)}^{2}} \end{array}$$

The UAV of Figure 3 consists of four brushless DC (BLDC) motors controlled by the onboard electronic speed controllers (ESCs) of the Toothpick F722 flight controller. The STM32F722 MCU on the flight controller board contains modified firmware to send out relevant data through the telemetry output via a Bluetooth point-to-point connection. A Python script receives the Bluetooth messages and stores them to estimate constants in postprocessing. The simplex communication from UAV to a computer includes information on battery voltage, power consumption, flight altitudes, vertical and horizontal speed, accelerations in XYZ directions, the angle of attack, yaw–pitch–roll angles, etc.

Depending on the environment, flight times can differ significantly. For example, less lift is produced for the same amount of energy in hot air because it is less dense than cold air. Similarly, lift decreases when increasing in altitude, or if humidity increases. The type of flying will also impact the flight time. Flight control algorithms which let the UAV fly as smoothly as possible are recommended, as they will yield the longest flight time.

#### 2.4.1. Completing the Proposed Model

To derive the model constants $\left\{k_{1},\;k_{2}\right\}$ and $\left\{c_{1},\;c_{2},\;c_{3},\;c_{4}\right\}$ from Equations (Equation 2) to (Equation 5), a total of three experiments in steady-state are required (the constant $c_{1}$ is defined as $\frac{k_{1}}{k_{2}}$).

##### Experimental Measurement 1: Hovering UAV with Payloads

First, a hovering experiment with different calibrated payloads $m_{\mathrm{payload}}$ is conducted, resulting in different average power consumption. Both the horizontal ($V_{\mathrm{hor}}$) and vertical speeds ($V_{\mathrm{vert}}$) are zero, thus the relation between power and weight during hovering can be written as Equation (Equation 6).
(6)$$P_{\mathrm{uav},\mathrm{hover}}=P_{\mathrm{avio}}+\left(c_{1}+c_{2}\right){\left(mg\right)}^{3/2}$$

The measured avionic power ($P_{\mathrm{avio}}$) (i.e., power when UAV is active but not flying) equals 10.6 W. The data points are scattered, as shown in Figure 4a, and the relations for the constants are estimated by the least squares (LS) fit via Equation (Equation 6) and gives $c_{1}+c_{2}=12.88$.

##### Experimental Measurement 2: Vertical Movements

Second, the power consumption was measured and averaged during ascent and descent at a constant vertical speed to heights of up to 50 m. Multiple repeated descents and ascents yielded four data points, where the vertical velocity $V_{\mathrm{vert}}$ is constant. Cosα and accelerations in the horizontal UAV axes were not allowed in these measurements, meaning that the horizontal speed ($V_{\mathrm{hor}}$) equals zero, and so the thrust *T* equals mg. Equation (Equation 7) shows the corresponding formula for the vertical power. Here, $c_{2}$ can be replaced by $12.88-c_{1}$ and $k_{1}$ by $k_{2}\;\cdot \;c_{1}$. Table 1 contains the parameters extracted from the LS fitting. The fitting was performed with five data points, two data points for ascending and two for descending, along with the energy consumption during hovering.
(7)$$P_{\mathrm{uav},\mathrm{vertical}}=k_{1}mg\left[\frac{V_{\mathrm{vert}}}{2}+\sqrt{{\left(\frac{V_{\mathrm{vert}}}{2}\right)}^{2}+\frac{mg}{k_{2}^{2}}}\right]+c_{2}{\left(mg\right)}^{3/2}$$

##### Experimental Measurement 3: Steady-Level Flight

Last, to calculate flight forward power, the UAV is flown forward at a number of constant speeds. Data points taken during high accelerations (i.e., at speed transitions) are neglected. Figure 4b shows the measurement data and corresponding least squares fitting. Equation (Equation 8) depicts the resulting formula. Since no useful angle of attack data could be extracted from the UAV measurements, cosα is assumed equal to 1. By fitting our model parameters to the resulting data, the UAV power consumption model is considered accurate enough for our purposes. The fitting here includes the cosine dependency in the model parameters, more specifically $c_{3}$ and $c_{4}$ of Equation (Equation 8).

To improve model parameters, outliers were removed. The final model parameters derived for the tiny UAV focused on are summarized in Table 1.
(8)$$P_{\mathrm{uav},\mathrm{horizontal}}=\left(c_{1}+c_{2}\right){\left[\sqrt{{\left(mg-c_{4}{\left(V_{\mathrm{hor}}\cos\alpha \right)}^{2}\right)}^{2}+{\left(c_{3}V_{\mathrm{hor}}^{2}\right)}^{2}}\right]}^{3/2}+c_{3}V_{\mathrm{hor}}^{3}$$

#### 2.4.2. Optimal Energy Per Meter

In the proposed UAV model, an optimal propulsion speed exists to minimize the amount of energy per meter ($E_{pm}$). This phenomenon occurs as a large amount of hovering energy is supplemented by only a fraction of this hovering energy related to speed, and it is illustrated in Figure 5. In normal conditions, the mass is 170 g, and an optimal speed of 19.5 m/s will result in an $E_{pm}$ of 3.7 J/m. The most efficient flight forward speed will shift to higher values as the weight of the UAV increases. In the following Section 3.2 and Section 3.3, the optimal speed for the corresponding UAV weight is taken as actual value in the calculations.

#### 2.4.3. Travel Energy

The UAV flies to the location via a given path with a total energy consumption of $E_{\mathrm{uav},\mathrm{travel}}$. The path can be divided into ascent, forward flight, and descent, corresponding to energy quantities $E_{\mathrm{uav},\mathrm{ascent}}$, $E_{\mathrm{uav},\mathrm{ff}}$, and $E_{\mathrm{uav},\mathrm{descent}}$, respectively. Before the horizontal movement is initiated, the UAV rises at a velocity $V_{\mathrm{vert}}$ to a predefined altitude $h_{1}$, it then flies horizontally to the location to subsequently descend to an altitude $h_{2}$. During horizontal displacements, the equations assume the shortest distance to move from the initial location to the IoT node at a horizontal velocity $V_{\mathrm{hor}}$ over a distance *d*. The second vertical displacement to the nodes location at height $h_{2}$ amounts to $h_{1}-h_{2}$. Moreover, the total mass of the UAV *m* is included in the travel energy. The formula accounting for the different parts of the path is shown in Equation (Equation 9) and uses the formulas of the UAV model from Section 2.4.
(9)$$\begin{array}{c} E_{\mathrm{uav},\mathrm{travel}}\left(h_{1},h_{2},V_{\mathrm{vert}},d,V_{\mathrm{hor}},m\right)=\underset{E_{\mathrm{uav},\mathrm{ascent}}}{\underset{︸}{P_{\mathrm{uav}}\left(V_{\mathrm{vert}},m\right)\;\cdot \;\frac{h_{1}}{V_{\mathrm{vert}}}}}+\underset{E_{\mathrm{uav},\mathrm{ff}}}{\underset{︸}{P_{\mathrm{uav}}\left(V_{\mathrm{hor}},m\right)\;\cdot \;\frac{d}{V_{\mathrm{hor}}}}}\\ +\underset{E_{\mathrm{uav},\mathrm{descent}}}{\underset{︸}{P_{\mathrm{uav}}\left(-V_{\mathrm{vert}},m\right)\;\cdot \;\frac{h_{1}-h_{2}}{V_{\mathrm{vert}}}}} \end{array}$$

## 3. Models for Energy Provisioning

The scenarios explained above are elaborated here with the aim of indicating how much energy is required under different conditions, such as energy needed on the IoT side, distance to the base station, etc.

### 3.1. Powering IoT Devices with RFPT

An RF transmitter sends electromagnetic energy as radio waves to the receiver antenna. The received power is converted to electrical energy. The system successively contains, from transmitter to receiver (from left to right in Figure 6), a waveform generator, a pa, an impedance matching circuit, two antennas, a secondary tuning network, a rectifier, a charge pump, and most often a maximum power point tracking (MPPT) DC/DC converter. Each element causes losses, which are included in the analysis and depicted in Figure 6.

Equation (Equation 10) combines all efficiency contributions to the total RFPT efficiency $\eta_{\mathrm{rfpt}}$, with $\eta_{\mathrm{pa}}$ the pa efficiency, $\eta_{t}$ the transmit antenna efficiency, $\eta_{w}$ the wireless link efficiency, $\eta_{r}$ the receive antenna efficiency, $\eta_{\mathrm{rec}}$ the rectifier efficiency, and $\eta_{\mathrm{reg}}$ the DC/DC conversion efficiency. The wireless link efficiency includes path loss, antenna gains or directivity, and potential polarization mismatch losses. Typically, the path losses are responsible for the highest loss contribution in a radio frequency power transfer system.
(10)$$\eta_{\mathrm{rfpt}}=\eta_{\mathrm{pa}}\;\cdot \;\eta_{t}\;\cdot \;\eta_{w}\;\cdot \;\eta_{r}\;\cdot \;\eta_{\mathrm{rec}}\;\cdot \;\eta_{\mathrm{reg}}$$

#### 3.1.1. Required Effective Isotropic Radiated Power

An expression is needed for the required EIRP for a given distance. Since EIRP represents the power just after the transmit antenna, the $\eta_{\mathrm{pa}}$ and $\eta_{t}$ from Equation 10 can be temporarily omitted and briefly addressed in Section 3.1.3. Expressions for the remaining efficiencies $\eta_{w}$, $\eta_{r}$, $\eta_{\mathrm{rec}}$, and $\eta_{\mathrm{reg}}$ are needed in this elaboration. The wireless link efficiency, $\eta_{w}$, mainly depends on the distance and the carrier frequency. The reflection coefficient of the receiver antenna is given by $\eta_{r}$. This antenna delivers RF energy to the harvester circuit. The harvester is responsible for converting the RF energy to direct current (DC) energy, where $\eta_{r}$ and $\eta_{\mathrm{rec}}$ depend to some extent on the amount of incoming RF power. To find a relation between transmitted RF power and received DC power, some assumptions are made. In this analysis, we consider a SISO system without beamsteering or beamforming capabilities.

The transmission formula of Friis, depicted in Equation (Equation 11), gives the expressions for efficiency parameters $\eta_{t}$, $\eta_{w}$, and $\eta_{r}$, with $|\Gamma_{t}|$ and $|\Gamma_{r}|$ the matching loss at transmitter and receiver, respectively, *d* the distance between them, $G_{t}$ and $G_{r}$ the antenna gains, and λ the wavelength. Thus, $P_{t}\;\cdot \;G_{t}\;\cdot \;\left(1-|\Gamma_{t}{|}^{2}\right)\;\cdot \;{\left|\rho_{t}\right|}^{2}$ is equal to the EIRP. Consequently, the received power $P_{r}$ [dBm] depends on the distance, wavelength, EIRP, and the receiver antenna gain, as formulated in Equation (Equation 12), expressed as dB-values. Furthermore, this assessment assumes no polarization losses $\rho_{t}=\rho_{r}=1$ and no matching losses $|\Gamma_{t}\left|=\right|\Gamma_{r}|=0$.
(11)$$\frac{P_{r}}{P_{t}}=\underset{\mathrm{TX}\;\mathrm{matching}\;\eta_{t}}{\underset{︸}{\left(1-|\Gamma_{t}{|}^{2}\right)}}\underset{\mathrm{RX}\;\mathrm{matching}\;\eta_{r}}{\underset{︸}{\left(1-|\Gamma_{r}{|}^{2}\right)}}\;\underset{\mathrm{Path}\;\mathrm{loss}\eta_{w}}{\underset{︸}{{\left(\frac{\lambda }{4\;\pi \;d}\right)}^{2}\;G_{t}\;G_{r}\overset{\mathrm{Polarization}\;\mathrm{loss}}{\overset{︷}{{\left|\rho_{t}\;\cdot \;\rho_{r}\right|}^{2}}}}}$$
(12)$$P_{r}=EIRP+G_{r}-20\log\left(\frac{4\pi d}{\lambda }\right)$$

To obtain a realistic estimate of the remaining conversion losses, a state-of-the-art energy harvester circuit is selected. The AEM40940 evaluation module can provide efficient DC power based on fluctuating input power using an MPPT algorithm. In summary, it can charge batteries and capacitors with RF energy and integrates a matching circuit, rectifier, and DC/DC converter [22].

The datasheet of the AEM40940 RF energy harvester provides a comprehensive discussion on the features of this IC. In addition, it includes performance curves relating the overall efficiency to the incoming RF energy. The overall efficiency is defined as the ratio of the DC power available at the output of the internal boost converter (set to 4.5 V) and the RF power at the input of the matching circuit ($P_{\mathrm{in}}$). The data are presented for multiple operating frequencies (867 MHz, 921 MHz, and 2.4 GHz), extracted from the datasheet [22] and transformed to the relation between the input RF energy and the DC output power ($P_{\mathrm{dc},\mathrm{out}}$). The raw data points and the fitted relation are shown in Figure 7.

A third-degree polynomial fitting, as formulated in Equation (Equation 13), results in values $a=8.45\times 10^{-5}$, $b=3\times 10^{-3}$, $c=3.84\times 10^{-2}$, and $d=2\times 10^{-1}$.
(13)$$P_{\mathrm{dc},\mathrm{out}}\left[mW\right]=a\;\cdot \;P_{\mathrm{in}}^{3}+b\;\cdot \;P_{\mathrm{in}}^{2}+c\;\cdot \;P_{\mathrm{in}}+d$$

We calculated the required EIRP as a function of distance and delivered DC power by combining Equations (Equation 12) and (Equation 13) with $P_{in}=P_{r}$. Figure 8 shows the relationship for output powers 10, 50, 100, 500 uW, and 1 mW. A working frequency of 868 MHz and half-wave dipole receiver antenna with a gain of 2.15 dBi are assumed. The former frequency was chosen due to its lower path loss compared with 2.4 GHz and because of fewer regulatory limitations on emitted power.

#### 3.1.2. Regulatory Restrictions

The harmonized ETSI standard *ETSI EN 302 208* discusses the radio frequency identification equipment restrictions. The equipment can operate in the 865 MHz to 868 MHz band with power levels up to 2 W and in the 915 MHz to 921 MHz band with power levels up to 4 W [5]. The levels, detailed in the standard, are expressed in effective radiated power (ERP). When using a half-wave dipole antenna, the EIRP is 2.15 dB higher than the corresponding ERP value. In conclusion, 38.15 dBm EIRP is the maximum power level allowed that can be lawfully emitted in this Industrial, Scientific and Medical (ISM) band. This limits the maximum distance when a certain DC power is required at the receiver. Figure 8 depicts the required EIRP power as a function of the distance and Table 2 shows the maximum distances for varying receive powers.

Techniques such as beamforming give a more directive beam but will not directly increase the operating distance. The reason is that the EIRP limit still applies. However, it is important to note that the antenna beamwidth can affect the EIRP values. As stated in [5], keeping the beamwidth within certain limits, results in higher permissible transmit power. The antenna beamwidth can be shaped by selecting patch antennas or implementing beamforming with multiple antennas.

#### 3.1.3. Power Amplifier

The preceding analysis omitted the pa losses. This component has a significant contribution to the total RFPT losses and should be included in efficiency analyses. Power amplifiers ensure amplified modulated signals. These circuits are connected to, and matched with, the transmit antenna. RFPT requires only an amplified single-tone signal, without modulated signals, connected to the transmit antenna. An amplified high-frequency sine wave can be generated via different circuits, each coming with their corresponding maximum achievable efficiencies.

Three types of efficiencies for power amplifiers are used in the literature [23]: the drain efficiency (DE), power-added efficiency (PAE) and $\eta_{\mathrm{overall}}$. In this paper, the drain efficiency DE suffices, meaning the RF output power $P_{\mathrm{RF},\mathrm{out}}$ divided by the input DC power $P_{\mathrm{DC},\mathrm{in}}$, represented in Equation (Equation 14), also denoted as PA efficiency ($\eta_{\mathrm{pa}}$).
(14)$$\eta_{\mathrm{pa}}=DE=\frac{P_{\mathrm{RF},\mathrm{out}}}{P_{\mathrm{DC},\mathrm{in}}}$$

The overall RFPT efficiency is further formulated in Section 4.1.1.

### 3.2. Energy Delivery to Battery-Powered Devices

A UAV can be used to provide power to IoT nodes. The total UAV power consumption is primarily determined by three factors:1.UAV flight and avionics consumption is a combination of hovering, ascending, descending, and steady-level flight energy. The model for the UAV is detailed in Section 2.4. The total energy consumption to fly to the node’s location and vice versa is called the total travel energy and is given in Section 2.4.3.2.The selected energy storage at the IoT node determines the charge rate and the amount of power transmitted to the node. A brief overview is given in Section 3.2.1.3.Power transfer technology between the UAV and IoT node imposes additional constraints on the system, such as limited distance between WPT-transmitter and receiver, efficiency, losses, and maximum transferable power, as discussed in Section 3.2.2.

#### 3.2.1. Energy Storage Selection

In applications, an IoT node is typically designed with a certain autonomy in mind. Even if a UAV-based approach is pursued, it is not practically feasible that batteries need to be recharged too frequently. Therefore, the storage technology must have a certain capacity to last several weeks/months on a single charge, and the battery charge time should be as short as possible, since the UAV is hovering in the meantime. In this article, three different types of rechargeable battery compositions are suggested for use in embedded devices. The three storage solutions, i.e., electrostatic double-layer capacitors (EDLC), lithium titanate (LTO), and lithium-ion capacitor (LIC), are compared in the spin diagram in Figure 9. These EDLC, LIC, and LTO cells can be charged much faster than conventional lithium cells, for which lithium cobalt oxide (LCO) is the most common composition [24,25]. This is a critical aspect in the proposed approach to avoid long hovering times.

One of the most crucial parameters during the selection of the storage solution is the self-discharge. We are carrying out a long-term experiment tracking self-discharge, as depicted in Figure 10. The results clearly indicate that EDLC cells have an excessive self-discharge rate. They are therefore not considered further.

The remaining options are further compared. Figure 9 shows that LTO or LIC chemistries come with reasonable volumetric energy densities while having very low internal resistance and high charge rates. Both technologies were charged at the maximum speed as advised in the manual and represented in Figure 11. An LTO battery with a measured capacity (KORAD KEL103) of 66.3 mAh is charged according to a constant current (CC) constant voltage (CV) method. With a 663 mA (10 C) charge current and a 2.8 V maximal voltage, the battery state of charge (SoC) rises to 90% after 6 min, corresponding with 581 J stored energy. Similarly, the charging process of the LIC cell with a capacity of 50 F was measured. With a charging current of 2.8 A and a maximum voltage of 4.2 V [26], it took approximately 2.6 to achieve a SoC of 90%, corresponding with 472 J stored energy. The charge power of the LTO cell is more constant in time compared with the LIC cell. LIC cells initially charge faster, resulting in the need for high current peaks. Charging components have to be dimensioned according to the higher peak current, as possible, with the fast-charging LIC cells. This can result in a higher bill of materials (BOM).

We further define the SoC at the start of the charging process as $SoC_{\mathrm{initial}}$ and the SoC at which charging is terminated as $SoC_{\mathrm{target}}$. A cut-off current, as defined in the manual, is used to determine the end of the charging process, i.e., at a 100% SoC. The time to achieve a fully recharged cell is rather high compared with the time to achieve a 90% SoC, since the amount of stored energy per unit time decreases when approaching a 100% SoC. Figure 11 also depicts this charging rate decrease.

Furthermore, the initial or remaining SoC has a positive impact on charge time but is not advantageous to the amount of transferred energy. For example, for the LTO cell, the time in CC-mode will decrease if $SoC_{\mathrm{initial}}>$ 0%. The average amount of stored energy per unit time will become lower. In short, it is better that the IoT cell is sufficiently discharged at the time the UAV comes to deliver energy.

LTO technology is further assumed to be implemented in the IoT devices for the following key reasons: the fast charge time in combination with the low self-discharge and the relatively high energy density.

#### 3.2.2. Energy Transfer Link

The survey in [4] summarizes the wireless power transmission technologies, subdivided in coupled and uncoupled approaches. The latter allows higher operating distances compared with the coupled technologies. Due to restrictions in radiation levels of both light and RF sources, LPT and RFPT systems are not suitable to quickly recharge the IoT energy storage. In addition to uncoupled technologies, the coupled approaches such as capacitive power transfer (CPT) or inductive power transfer (IPT) need a very strict alignment method. In the scenarios considered in this article, strong alignment is difficult to achieve, and thus the CPT and strongly coupled IPT systems are not good choices to transfer the energy. An inductive system with both receiver and transmitter in resonance, i.e., Magnetic Resonance Coupling (MRC), provides the most appropriate solution. IPT systems operate with highly coupled and aligned coils (coupling factor *k* > 0.3 [27]). MRC systems, on the other hand, can operate on coupling factors below 0.1, giving these resonant systems more spatial freedom and better handling of fluctuations in *k* values. MRC is based on two coils tuned to the same resonance frequency, in which, even at low coupling factors, high link efficiencies can be achieved. The amount of power required through the link will range from a few 100 mW to several watts, which is transferable with MRC systems.

To accomplish further analysis in Section 3.2.3, an assessment of the efficiency of the WPT system is performed. The transmitter PCB coil fits with the proposed UAV frame of Figure 3 with dimensions 145 by 165 mm and a total of two windings. The receiver PCB coil has four windings and dimensions of 100 by 100 mm. Via Ansys Maxwell finite element analysis, self-inductances ($L_{1}L_{2}$), series resistances ($R_{1}R_{2}$), and mutual inductance (*M*) between the coils are obtained. The configured frequency is set to 6.78 MHz, as also used by the Airfuel Alliance Resonance [28]. The coupling factor ($k_{\mathrm{mrc}}$) is calculated with Equation (Equation 15).
(15)$$k_{\mathrm{mrc}}=\frac{M}{\sqrt{L_{1}+L_{2}}}$$

Based on the coupling factor, the maximum link efficiency ($\eta_{\mathrm{mrc}}$) can be calculated using Equation (Equation 16) [29], where $Q_{T}$ and $Q_{R}$ are the quality factors of the transmit and receive coils, respectively. This assumes the most optimal load on the receiver side. If the receiver circuit is not loaded with the optimal resistance, the link efficiency decreases. Additional analysis assumes an optimal load.
(16)$$\eta_{\mathrm{mrc}}=\frac{k_{\mathrm{mrc}}^{2}Q_{T}Q_{R}}{1+k_{\mathrm{mrc}}^{2}Q_{T}Q_{R}}$$

We conducted finite element simulations for two vertical misalignments of 100 mm and 150 mm. Lateral misalignment is being neglected, and angular misalignment is minimal, because the UAV coil is located parallel to the receiver coil during hovering at the node’s location. The results are shown in Table 3.

The simulations show that the maximum link efficiency will achieve percentages of 50% or more for the selected coils at distances of up to 150 mm. Additionally, the total efficiency of the WPT system ($\eta_{\mathrm{wpt}}$) is composed of several other individual component contributions, such as the preamplifier ($\eta_{\mathrm{pre}-\mathrm{amp}}$), power inverter ($\eta_{\mathrm{inv}}$), rectifier ($\eta_{\mathrm{rect}}$), and battery charger ($\eta_{\mathrm{smps}}$). The $\eta_{\mathrm{wpt}}$ efficiency is stated in Equation (Equation 17).
(17)$$\eta_{\mathrm{wpt}}=\eta_{\mathrm{pre}-\mathrm{amp}}\;\cdot \;\eta_{\mathrm{inv}}\;\cdot \;\eta_{\mathrm{mrc}}\;\cdot \;\eta_{\mathrm{rect}}\;\cdot \;\eta_{\mathrm{smps}}$$

It would take the reader too long to elaborate on all components of an MRC system individually. We assume low losses in the remaining components. Due to new technologies such as GaN FETs, high efficiencies (>0.9) in DC/AC inverters and DC/DC converters are feasible. In the further analyses, we assume a total efficiency $\eta_{\mathrm{wpt}}$ of 50%. While we so far only discussed the efficiency of a WPT system, evidently, limitations in energy transfer due to component selection also exist.

#### 3.2.3. Limitations of IoT Storage Capacity

A first analysis investigates the maximum size of the IoT storage capacity. The capacity cannot be selected as infinitely large, since the UAV would have to hover for a very long time to transfer energy to the node. Furthermore, the available energy of the UAV is limited to the selected UAV battery capacity, which consequently translates to limitations in the size of the IoT storage capacity. For the IoT storage buffer, we assume a 90% charge after each intervention. The nodes are reloaded from an initial SoC of 0% to the target SoC of 90%.

The analysis is performed for the UAV in Figure 3 weighing 170 g with a battery capacity of 9.62 Wh. The distance between departure and arrival location is given by *d* and can be gradually increased to test the limits of this setup. The optimal speed from Section 2.4.2, here named $V_{\mathrm{hor}}$, is 19.5 m/s. It is further assumed that the UAV flies to 50 m ($h_{1}$) altitude at the departure location and hovers to 10 m ($h_{2}$) altitude at arrival. In both cases, the vertical speed is set at 10 m/s ($V_{\mathrm{vert}}$). These proposed parameters allow the UAV travel energy, explained in Section 2.4.3, to be calculated.

In this example, an LTO cell is charged at the node’s location. We assume, the charge process follows a simplified linear model. Mei et al. [30] already measured the charge time for LTO cells related to different C-rates as being the rate at which a battery can be charged relative to its capacity, specifically in CC-mode. The charging time (in CC-mode) decreases from 27.5 min to approximately 6 min when changing the charge rate from 2 C to 8 C, respectively. As shown in Figure 11, the charging curve proceeds linearly to about 90% SoC. In our measurement, the maximum C-rate is set to 10 C. The maximum charge rate may vary depending on the brand of LTO cell selected [31,32].

We assume a UAV equipped with an inductive transmitter system with an efficiency of 50%, as explained in Section 3.2.2. The WPT link is limited to a certain transfer power $P_{\mathrm{ch},\max}$. This analysis assumes a received power of 1 W. Based on the received power, the C-rate can be determined via Equation (Equation 18) together with the battery capacity $E_{\mathrm{node}}$. The analysis additionally limits the maximal C-rate to 10 C.
(18)$$C_{\mathrm{rate}}=\frac{P_{\mathrm{ch},\max}}{E_{\mathrm{node}}}\;\cdot \;3600\leq 10\;C$$

Based on the parameters such as the output power sourced from the WPT receiver, the target SoC, and lastly the size of the IoT storage, an estimate of the charge time $t_{ch}$ can be found according to Equation (Equation 19).
(19)$$t_{ch}=\frac{3600}{C_{\mathrm{rate}}}\;\cdot \;\left(SoC_{\mathrm{target}}-SoC_{\mathrm{initial}}\right)$$

The total energy $E_{\mathrm{uav},\mathrm{overall}}$ required for an intervention can be calculated with Equation (Equation 20) and is subdivided into three parts:1.The travel energy $E_{\mathrm{uav},\mathrm{travel}}$ from Equation (Equation 9) for the outward and return flights. We multiply the traveling energy by two. Although, if $h_{1}$ and $h_{2}$ are different, there is a small deviation between the outward and return travel energy present. This small deviation is neglected here.2.The hovering energy $E_{\mathrm{uav},\mathrm{hover}}$, which depends on the charge time and hovering power from Equation (Equation 6).3.The total amount of energy consumed by the WPT transmitter $E_{\mathrm{wpt},\mathrm{transmitter}}$, with $\eta_{\mathrm{wpt}}$ the efficiency determined by the energy delivered by the UAV battery relative to the energy stored in the node.
(20)$$E_{\mathrm{uav},\mathrm{overall}}=2\;\cdot \;E_{\mathrm{uav},\mathrm{travel}}\left(h_{1},h_{2},v_{v},d,v_{h},m\right)+\underset{E_{\mathrm{uav},\mathrm{hover}}}{\underset{︸}{t_{ch}\;\cdot \;P_{\mathrm{uav},\mathrm{hover}}}}+\underset{E_{\mathrm{wpt},\mathrm{transmitter}}}{\underset{︸}{{\left(\eta_{\mathrm{wpt}}\right)}^{-1}\;\cdot \;E_{\mathrm{IoT},\mathrm{storage}}}}$$

Figure 12 shows the total energy requirement related to the size of the IoT battery for multiple distances. Taking into account the considered boundary conditions, the selected UAV will be able to recharge a storage of ≈ 900 J over a distance of 500 m.

The slope remains rather flat for smaller IoT batteries, due to the charge rate limitation of the chosen battery composition, in this case, 10 C for the LTO cell. A lower travel distance logically provides the opportunity to transfer larger amounts of energy and consequently store higher quantities of energy. In contrary, at increased distances, limitations in storage sizes will arise due to UAV battery capacity limits, indicated by the dotted line. Based on this graph and the daily consumption of the application, the autonomy of the IoT device can be determined. Subsequently, the number of yearly interventions can be calculated.

#### 3.2.4. Duration of an Intervention

Figure 13 gives an estimate of the total intervention duration. The calculation accounts for the UAV battery capacity limitations, resulting in finite curves when battery capacity is insufficient. Remarkably, longer fly times can occur for shorter travel distances (e.g., d= 0.5 km) with the same UAV battery capacity. In that case, most of the intervention time is attributable to the node’s charging process and not due to travel time, since this is constant for each distance. This analysis was performed under the same conditions used in Figure 12.

### 3.3. Swapping Embedded Batteries

In this third scenario, we consider a UAV capable of replacing the IoT node’s battery. Rechargeable battery compositions make sense to use for environmental reasons, as elaborated in Section 5. non-rechargeable batteries can also be used for battery swapping, although their lifespan is much shorter than that of rechargeable batteries, which can be reused many times. For the actual energy calculations in this analysis, it is irrelevant whether the battery is rechargeable or not. The properties of the selected battery composition are more important.

Batteries with high weight energy densities are recommended here. These battery types are usually charged by means of lower C-rates than, e.g., the already discussed LTO, LIC or EDLC compositions. These higher storage capacities can, on the one hand, enable applications with higher energy needs or, on the other hand, increase autonomy. The latter consequently also reduces the number of UAV interventions. The drained batteries can be recharged at slow C-rates in the same place as the UAV battery, since there is no need for the recharging to happen on-site. Table 4 lists some commonly used batteries and shows the nickel cobalt aluminum (NCA) battery composition will provide the best weight energy density.

Other important parameters, such as self-discharge, sustainability aspects, life cycle assessment (LCA) of the batteries, shelf life, etc., are neglected here. These factors may impact the rudimentary choice to select for the highest weight energy density. Section 5 discusses briefly the environmental impacts of these different battery types.

This approach poses mainly practical implementation challenges, such as the implementation and design of the swapping mechanism. Existing work has already investigated battery-swapping mechanisms, more specifically, the replacement of a UAV battery [38,39]. Similar implementations can be explored to replace the IoT node’s battery. Ultimately, this process must take place autonomously, whereby the efficiency highly depends on the replacement time ($t_{\mathrm{replacement}}$), i.e., the time to complete the battery-swapping process. The battery should be easily detachable from the system or perhaps connected contactlessly to provide a smooth replacement. For the purpose of this paper, we neglect the mechanical difficulties of the battery-swapping system and focus on the energy reloading capabilities and efficiency.

The previous Equation (Equation 20) can be adapted to Equation (Equation 21) to estimate the overall energy consumption in the battery-swapping operation. Note that the added weight of the battery ($m_{\mathrm{payload}}$) to the UAV weight ($m_{\mathrm{uav}}$) will have a significant impact on the results. Several assumptions are made to estimate the UAV power consumption. The model from Section 2.4 is used, and the UAV is considered to travel at the weight-dependent optimal speed, as calculated in Section 2.4.3. A very cautious estimate of 60 s, i.e., $t_{\mathrm{replacement}}$, for a battery swap is assumed. In terms of weight, both outbound and return flights carry the same payload.
(21)$$\begin{array}{c} E_{\mathrm{uav},\mathrm{overall}}=2\;\cdot \;E_{\mathrm{uav},\mathrm{travel}}\left(h_{1},h_{2},v_{v},d,v_{h,optimal}\left(m_{\mathrm{uav}}+m_{\mathrm{payload}}\right),m_{\mathrm{uav}},m_{\mathrm{payload}}\right)\\ +\underset{E_{\mathrm{uav},\mathrm{hover}}}{\underset{︸}{t_{\mathrm{replacement}}\;\cdot \;P_{\mathrm{uav},\mathrm{hover}}\left(m_{\mathrm{uav}}+m_{\mathrm{payload}}\right)}} \end{array}$$

Figure 14 depicts the UAV energy needs for varying payloads and servicing distances. The UAV can carry a maximal payload of 80 g, and the selected battery contains a maximum usable energy of 34.63 kJ. These restrictions are represented by the green shading in the figure. For example, taking into account the discussed assumptions with a 20 g IoT battery and a travel distance of 2 km, a UAV energy of approximately 20 kJ is required to perform the task.

## 4. Comparative Survey and Results

This section evaluates the three scenarios based on figure of merit (FoM) parameters, complemented by a comparative discussion. It further investigates whether the results can be improved by selecting other types of UAVs.

### 4.1. Figure of Merit Analysis

To provide a means to assess the advantages of the scenarios introduced in Section 3, this section makes a comparison and defines a FoM. We first briefly summarize how the scenarios differ from each other.

(S1)Section 2.1 describes an RFPT SISO system. Since this can theoretically be switched on indefinitely, and hence time is not an issue, the power consumption is the main focus. The efficiency $\eta_{\mathrm{rfpt}}$ is calculated through the relation between DC output power and DC input power.(S2)Section 3.2 rather concentrates on providing energy on site. The efficiency $\eta_{\mathrm{delivery}}$ is defined by the stored energy in the IoT device related to consumed UAV energy.(S3)The approach, explained in Section 3.3, requires another FoM calculation to evaluate the performance. The UAV with battery-swapping capabilities transports energy based on a separate energy storage medium, i.e., a reusable battery. The energy efficiency here no longer depends on how efficiently the UAV’s battery is used but rather on the amount of energy delivered compared with the amount of initial energy. This initial energy is made up of the combined capacities of the carried and UAV batteries.

#### 4.1.1. Radio Frequency Power Transfer Efficiency

The models from Section 2.1 defined the RF transmit power to get a certain amount of DC power on the receiver. Efficiency is quantified based on these estimates. In order to describe the RFPT overall efficiency, a realistic nonideal PA is assumed. Examples from Johansson and Fritzin [40] show that PAs can achieve drain efficiencies of up to 77% for power outputs above 31 dBm. The results depend on the distance and, to a lesser extent, the required receiving power. The efficiency is illustrated in Figure 15. It is important to notice that all efficiency values <0.1%, while only ranges lower than 5 m are considered for this scenario.

#### 4.1.2. UAV as Mobile Powerbank: Overall Efficiency

The overall efficiency $\eta_{\mathrm{uav},\mathrm{charge}}$ is a straightforward extension of the previous analysis and can be determined by Equation (Equation 22). $E_{\mathrm{uav},\mathrm{overall}}$ is elaborated in Section 3.2 and consists of the travel energy for outward and return flights, combined with the hovering energy while charging, as well as the energy consumed in the WPT transmitter. $E_{node}$ is the energy received at the IoT node.
(22)$$\eta_{\mathrm{uav},\mathrm{charge}}=\frac{E_{\mathrm{node}}}{E_{\mathrm{uav},\mathrm{overall}}}\;\cdot \;100\;\%$$

Figure 16 presents the overall efficiency related to the storage capacity of the IoT device. On the one hand, tiny IoT storage capacities logically result in lower efficiencies. On the other hand, oversized IoT storage capacities result in a situation in which no solution for the overall efficiency can be found. This is due to limitations in UAV battery capacity. The required energy (travel energy, hovering energy and UAV transmitter energy) exceeds the battery capacity, leading to an incalculable efficiency. Note that at the end of the curve, the UAV battery is fully utilized.

The dotted lines represent estimates of the UAV equipped with a dual battery. Obviously, this extra weight causes an increase in hovering energy. Therefore, the overall efficiency decreases for similar distances, despite the fact that the IoT storage capacities may increase. This is represented in the graphs of Figure 16.

A brief analysis of the parameter impact on overall efficiency is essential here. Consulting Equation (Equation 22) reveals that primarily $P_{\mathrm{ch},\max}$ strongly influences the overall efficiency. More specifically, $\eta_{\mathrm{uav},\mathrm{charge}}$ increases by a factor of ≈7, if the $P_{\mathrm{ch},\max}$ increases from 1 W to 10 W. In addition, variations in WPT efficiency could also influence the overall energy efficiency and show to have a higher impact when $P_{\mathrm{ch},\max}$ is large. For example, the variation from 10% to 50% WPT maximum achievable efficiency at $P_{\mathrm{ch},\max}$ of 1 W and a travel distance of 1000 m indicate an increase from 1.7% ($E_{\mathrm{node}}=$ 615 J) to 2.1% ($E_{\mathrm{node}}=$ 754 J). A $P_{\mathrm{ch},\max}$ of 10 W for similar travel distance provides an increase in $\eta_{overall}$ from 4.2% ($E_{\mathrm{node}}=$ 1487 J) to 14.1% ($E_{\mathrm{node}}=$ 5005 J).

#### 4.1.3. Swapping IoT Batteries: Overall Efficiency

To evaluate this third scenario, the efficiency ($\eta_{\mathrm{uav},\mathrm{swap}}$) in Equation (Equation 23) is determined here, now in a slightly different way. Since the delivered energy is not extracted from the UAV battery, the carried energy must be included in the denominator. $E_{\mathrm{uav},\mathrm{overall}}$ equals Equation (Equation 21), which amounts to two times the travel energy, along with the hover energy during the swapping process.
(23)$$\eta_{\mathrm{uav},\mathrm{swap}}=\frac{E_{\mathrm{node}}}{E_{\mathrm{uav},\mathrm{overall}}+E_{\mathrm{node}}}\;\cdot \;100\;\%$$

The efficiency is a function of the payload and depends heavily on the weight energy density of the battery technology. An NCA battery is assumed to be swapped in this graph. As shown in Figure 17, the curves increase when heavier batteries are selected. It is unrealistic to assume that the payload exceeds the weight of the UAV multiple times. As mentioned earlier, we consider a UAV of 250 g, implying a residual maximal payload of 80 g. Figure 17b shows the adapted curves, when the UAV’s battery is doubled. In the latter case, the payload is limited to 21 g.

### 4.2. Figure of Merit Comparison and Overall Discussion

As earlier demonstrated in Section 4.1.1, the RFPT system reaches very low efficiency values, i.e., of a few hundredths of a percent and for distances in the range of meters. This tremendous disadvantage means that this technology has very limited use cases, i.e., only usable in very low-power applications. The overall advantage of RFPT is the nearly infinite amount of time that small amounts of energy can be delivered. Furthermore, this technology can serve multiple devices simultaneously, in contrast to the UAV approaches.

Unlike RFPT, UAV-based servicing has clear advantages due to the large serviceable area. The disadvantage is the more complex implementation, requiring flight planning algorithms and battery-swapping mechanisms, posing wireless charging challenges, etc. This section primarily aims to evaluate efficiencies in various situations. It does not make sense to compare the short wireless power range with a possible long-range UAV solution. It is more meaningful to compare the charge and swap process efficiencies from Section 4.1.2 and Section 4.1.3 respectively and not further consider the RFPT technology here.

Table 5 compares several situations. Note that this comparison focuses on the side of the energy transfer to the node. The relation with parameters impacting the energy consumption of the node consumption, e.g., number and size of packets, is out of scope of this comparison. We refer to related work clarifying these aspects [2,10]. In all cases, the optimal speed criterium for the UAV, depicted in Section 2.4, is met. The battery technologies LTO and NCA at the IoT node are assumed for the charging and swapping process, respectively. The payload (including the optional second UAV battery) is still limited to 80 g. The maximum charge power is set to $P_{\mathrm{ch},\max}=$ 10 W. Based on these assumptions, along with the number of UAV batteries, travel distance, and IoT storage capacity, the FoM values are calculated. The UAV flying characteristics and weather conditions are assumed equal in the comparisons. The remaining $SoC_{\mathrm{uav}}$ was added to the table to make a fair comparison. Below, these four situations are briefly discussed.

(Cf. 1)Battery swapping, even with a full payload, is more efficient than in situ wireless battery recharging. In this case, the UAV consumption is even smaller than the transported energy. Figure 17 already showed that higher payloads lead to higher FoM values, although an amount of 68.3 kJ of rechargeable IoT storage is rather unrealistic for low-power IoT integrations. They do not benefit from an oversized battery, since this reduces the environmental friendliness of the approach. A more energy-efficient solution is to use the full payload capability, which means transporting multiple IoT batteries at the same time to serve more devices in a given area. This analysis is not covered in this study.(Cf. 2)An equally large IoT battery capacity of 5 kJ with a similar FoM demonstrates that the swapping process appears to be the most advantageous once again. In this case, the distance can reach 2.6 km further located nodes while maintaining the same FoM value.(Cf. 3)If both numbers of UAV batteries, travel distance, and storage capacity are assumed equal, the swapping process is still advantageous. Furthermore, the UAV battery may preserve a significant amount of residual energy.(Cf. 4)Continuing on (Cf. 3), the difference in FoM values becomes smaller with increasing distance. In this example, there is only a 2.5% difference. Obviously, the remaining SoC is higher when swapping, since the charge energy comes from the UAV battery.

Another way to compare wireless charging with battery swapping is to plot the FoM with respect to the distance. As shown in the previous analysis from Table 5, the UAV battery is not necessarily fully drained after an intervention. To give a better picture, both efficiency and the remaining $SoC_{\mathrm{uav}}$ are plotted in Figure 18. In both cases, it is assumed to bring an energy quantity of 1000 J to the IoT device location. The remaining assumptions are summarized in Table 6, with the UAV battery that is still 9.62 Wh and the hovering time $t_{\mathrm{hover}}$ equal to $t_{\mathrm{ch}}$ or $t_{\mathrm{replacement}}$ for charging or swapping processes, respectively. Furthermore, the parameters $P_{\mathrm{ch},\max}$, $C_{\mathrm{rate}}$, and $\eta_{\mathrm{wpt}}$ remain crucial for estimating the energy consumption of the charging approach. Moreover, the weight energy density of the carried battery is essential to estimate the intervention energy for UAVs with swapping capabilities.

In addition to the fact that the swapping approach appears more efficient, this graph also represents the remaining energy of the UAV battery. In some cases, especially for closer located devices, the UAV could serve multiple IoT devices without recharging itself. This can be beneficial for the overall FoM value, since more energy is supplied to the drained IoT devices during one intervention. Additional analysis on serving multiple devices with energy is not elaborated on in this manuscript.

Previous assessments neglected the weight of the charging mechanism and the swapping mechanism. Results can differ when assuming a different hovering time for the battery replacement. The duration of 1 min to swap a battery is currently a rough estimation. Even when considering a battery-swapping time of 2 min, the swapping process is still advantageous, e.g., in (Cf. 1), the FoM becomes 66.7%, and in (Cf. 4), the FoM becomes 7.9%. In future work, the swapping process can be further studied, and the impact of mechanical options can be identified based on actual implementations.

### 4.3. UAV Optimizations

The UAV considered in this study is well-suited to give a proper estimate of the possibilities and challenges of UAV-based servicing of IoT nodes. Since this a very small UAV due to legislation restrictions, power consumption is quite low. There are more efficient UAVs models available, which could result in an even lower $E_{\mathrm{pm}}$. Larger blade spans combined with slower turning motors can yield a more efficient flying setup, but more optimal solutions exist based on propeller and motor selection [41]. The frame was chosen for its versatility and adaptability. A 3D-printed version of the frame was realized, which resulted in less than ideal flight characteristics. There were minor vibrations throughout the entire structure, since the frame was not as rigid as the carbon fiber version. Moving to a stiffer frame could greatly reduce vibrations. In a small UAV, the power consumption of the flight electronics has a significant impact on the total flight time, i.e., for this model 20% of the total power consumption. Lithium polymer (LiPo) batteries are typically used in UAVs due to their high discharge rate [42]. Lithium-ion (Li-ion) batteries can prolong flight times by offering higher energy-to-weight densities [33]. The relatively low discharge rate of Li-ions makes them only suitable in UAVs with a low power consumption to weight ratio.

## 5. Other Ecological Considerations and Future Work

Related studies, such as [43], only focus on energy consumption as an indicator of environmental impact. While this is an important metric, it misrepresents the true ecological impact of a particular device if no consideration is given to the environmental impact of the battery production and manufacturing of the UAV components. In this section, other ecological impacts to consider are introduced, and topics that require or deserve further research are summarized in the future work subsection.

The direct environmental implications of physical devices across their life cycle are frequently disregarded. LCA can here be a great tool for transitioning from a traditional cradle-to-gate to a cradle-to-grave analysis, accounting for all necessary steps while quantifying the true impact of IoT systems. A device’s environmental impact typically consists of two main contributions. First, the actual environmental impact of a particular product, for which the Global Warming Potential (GWP) metric is commonly used [44]. Second, the finite availability of resources, in particular resource depletion, which we further elaborate on in Table 7.

Only energy for operation has been taken into account so far in this work. While assessing the environmental footprint related to energy usage, the location of manufacturing and, consequently, the location of operation are also important to consider. The environmental impact will greatly differ due to the manner in which energy is generated. Energy from China (544 gCO2eq/kWh) or India (637 gCO2eq/kWh) has a significantly higher GWP than energy generated in Europe (mean 290 gCO2eq/kWh), e.g., Sweden (44 gCO2eq/kWh) [45].

An initial reflection is provided to better understand the ecological impact of a UAV-based recharging system. Since a complete LCA analysis is not the main purpose of this paper, this work is only a lead to create awareness for the reader that there is a plethora of aspects to consider for a sustainable IoT story, and many R&D questions are open in this domain. We are happy to refer the reader to upcoming papers concerning this topic in future work. Our UAV-based energy provisioning system contributes to a more ecological IoT in two ways. First, our approach reduces the ecological impact compared with previous fire-and-forget approaches. Second, we have the possibility of actively extending the battery life almost limitlessly.

### 5.1. Impact Comparison

A preliminary LCA analysis is conducted using publicly available carbon footprint data. Since GWP data for lithium-based batteries differ quite a lot in the literature, data from four sources [46,47,48,49] were collected and averaged, yielding a quite good estimate for lithium battery manufacturing GWP. For non-rechargeable (alkaline) batteries, data from [50,51] are used. The UAV model is based on a small sub- 250 g model from Pirson and Bol [44], assuming a typical lifetime of 400 flight hours [52], a power consumption of 50 W (resulting from Figure 4b), assuming European grid energy [53], and a flight time of 13 min (as depicted in Figure 13). The GWP of an extra four batteries, assuming a life cycle of 300 cycles, are included due to battery degradation during the lifetime of the UAV. An IoT energy consumption of 20 J a day for 10 years and a lithium-based rechargeable battery with a capacity of 2 kJ are assumed.

Figure 19 depicts the total impact of the UAV battery replacement compared with using a non-rechargeable battery. Only a single battery is assumed to be used during a 10 year lifetime of a single IoT node. Since a much smaller rechargeable than non-rechargeable battery can be used, the GWP thereof can be reduced significantly. Assuming a battery capacity of 2 kJ, a total of 36.5 UAV flights are needed over the course of 10 years to top up the battery. The lifetime of the UAV is assumed to be 400 flight hours (13 min/flight), incorporating the GWP of two battery replacements during its lifetime. The IoT node power consumption is assumed to be 20 J/day. The total GWP of the UAV-based servicing approach (in green) can be environmentally beneficial for powering the IoT node compared with using non-rechargeable batteries (in red). The total environmental impact of using non-rechargeable batteries comes down to 940 gCO2eq, while the UAV-based servicing approach yields a GWP of only 555 gCO2eq. An even lower GWP solution can be designed when optimizing the battery capacity at the IoT node and UAV, taking into account an optimal number of flights.

### 5.2. Scarcity of Elements

Only GWP data are available on the most commonly used battery types, e.g., lithium–polymer, lithium-ion, and alkaline batteries. On the GWP of certain battery types, such as LTO or LIC, as considered in this work, there is currently little to no information available. Therefore, we consider the scarcity of elements [54] needed for the manufacturing of the batteries as a way of qualifying the most environmentally friendly battery type. Table 7 gives an overview of the scarce elements required in the considered battery types.

Nickel manganese cobalt (NMC), NCA, LTO, and the older nikkel metal hydride (NiMH) and nikkel cadmium (NiCd) batteries are considered a bad environmental choice, since they use the most amount of scarce elements, i.e., one or two element(s) in the rising threat category and three or four elements in the future risk category. Lithium iron phosphate (LFP) is considered a better choice, since it uses no elements from the rising threat category and only two elements from the future risk category.

**Table 7 sensors-23-02291-t007:** Scarcity of elements used in battery production: elements with rising threat from increased use (orange), limited availability future risk to supply (yellow), and plentiful supply (green) [54,55,56,57,58,59].

		Cobalt	Cadmium	Chromium	Lithium	Nickel	Manganese	Phosphor	Vanadium	Zirconium	Fluor	Iron	Kalium	Aluminum	Titanium
Nickel manganese cobalt	NMC	x			x	x	x	x			x				
Lithium cobalt oxide	LCO	x			x			x			x				
Nickel cobalt aluminum	NCA	x			x	x		x			x			x	
Lithium iron phosphate	LFP				x			x			x	x			
Lithium titanate	LTO/NMC	x			x	x	x	x			x				x
Nikkel cadmium	NiCd	x	x			x							x		
Nikkel metal hydride	NiMH	x		x		x			x	x		x	x		x

### 5.3. Manual Labor Comparison

IoT system maintenance is currently performed manually, which requires a lot of human effort and will not be practical in the future due to the fast-growing number of edge devices [1]. Table 8 compares traditional manual maintenance scenarios with the automated approach presented in this work. The required energy per unit kilometer for each intervention type is estimated. The gasoline car results are based on available energy per gallon, assuming a typical efficiency of 20–25% [60]. For electric cars, the energy consumption can be directly used, assuming charging is 100% efficient. The amount of ‘food energy’ (kcal) is taken into account for walking and cycling. For the big UAV energy consumption, a 3 kg system is assumed [61], while for the small UAV, values from Figure 5 are used, assuming an $E_{\mathrm{pm}}$ of 5 J/m. A gasoline-based car uses the most energy per kilometer, i.e., 562 Wh/km. An important improvement can be made using small UAVs in autonomous servicing systems, consuming only 1.39 Wh/km. Additionally, fully automated servicing systems require no user intervention during the lifetime of the product, saving time and human effort, while improving energy efficiency.

### 5.4. Future Work

The analyses on the total UAV consumption in this paper are based on reasonable estimates. In future work, a validation, by means of actual measurements, would significantly improve current estimates. Local circumstances, such as weather, can lower WPT efficiency levels or lead to an increase in UAV power consumption. Additional elements contributing to the avionic power, e.g., a swapping mechanism, can also lead to reduced flight times. An extension of the concept from Section 3.3 does not need to be limited to just replacing batteries. Other IoT node parts, such as sensors, central motherboards, or even full devices can be replaced in case of malfunctioning or failure. Here, a new research topic on designing the next generation of serviceable IoT devices arises. This can be supplemented by research into biodegradable materials to implement certain IoT functions, only engaging the UAV for the retrieval of remaining harmful parts.

This work shows a UAV-based servicing approach has many benefits compared with manual replacement. It eliminates manual labor, reduces energy consumption, and lowers the GWP compared with using non-rechargeable batteries. Still, the full impact of the proposed concepts on the environment needs to be further evaluated, since here only an overview of the rare elements in batteries was elaborated. By reducing battery capacity, we can also lower the ecological impact of a typical system. However, this will have ecological repercussions due to the more frequently required UAV flights to deliver the same amount of energy. Hence, optimizations can be performed. It can also be examined whether the environmental impact of an IoT application is more likely to increase or decrease by implementing an additional harvester circuit. Furthermore, the environmental impact of the energy required to fly to the IoT node can be reassessed, taking into account that the share of renewable energy still increases annually.

## 6. Conclusions

In this work, we presented options, assessments, and potential optimizations of a unmanned aerial vehicle (UAV)-based servicing approach. We compared the radio frequency power transfer (RFPT) system with a UAV approach and clarified the convenient yet very low efficiency of RFPT in short-range energy transfers, while UAVs can achieve a much higher efficiency, and moreover, can be utilized for long-range energy provisioning. A UAV energy consumption model was derived to provide accurate estimates of flight energy consumption. The comparison of battery types at the Internet of Things (IoT) node in terms of energy density and ecological impact shows that lithium titanate (LTO) is a promising rechargeable battery technology for the on-site recharging approach, while nickel cobalt aluminum (NCA) has the highest energy density, which benefits the efficiency of the discussed swapping process. When examining UAV approaches, our results show that it is more energy efficient to swap instead of recharge the IoT battery. For the case of wireless in situ charging, an actual system implementation is feasible based on known designs. Conversely, autonomous replacement will require a more complex mechanical design.

This work shows a UAV-based servicing approach has many benefits compared with manual replacement. It eliminates manual labor, reduces energy consumption, and lowers the Global Warming Potential (GWP) compared with using non-rechargeable batteries.

We can conclude from the work presented in this paper that UAV-based servicing of IoT nodes clearly is of interest from an operational energy efficiency perspective. Moreover, other ecological benefits could be achieved, requiring dedicated study and opening many new research questions.

## Figures and Tables

**Figure 1 sensors-23-02291-f001:**
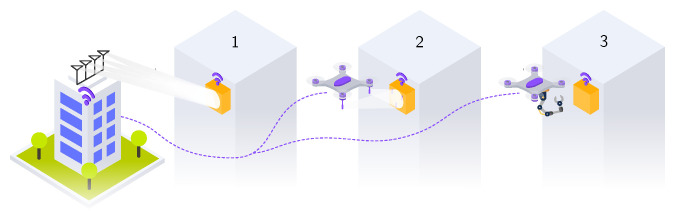
Overview of different solutions: (1) uncoupled wireless power transfer, (2) wireless charging by UAV equipped with WPT transmitter in proximity of the IoT node, and (3) battery replacement by UAV.

**Figure 2 sensors-23-02291-f002:**
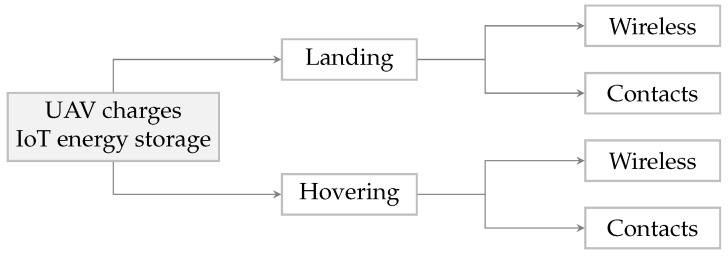
Overview of different subscenarios for IoT charging with UAV.

**Figure 3 sensors-23-02291-f003:**
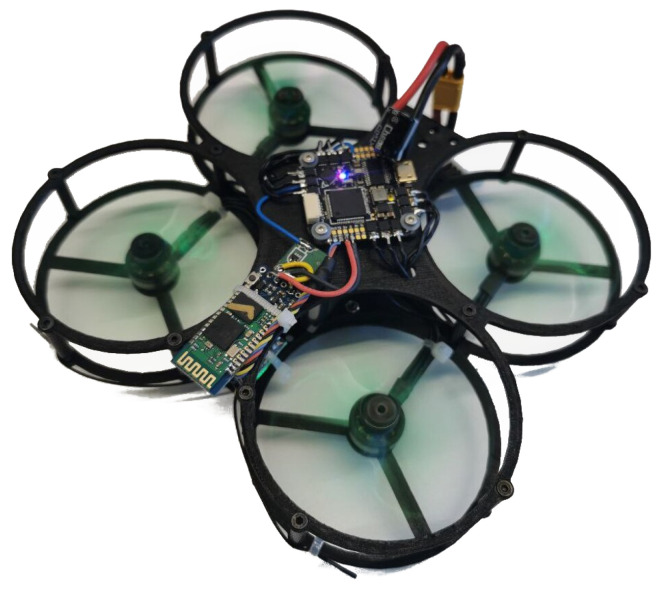
Picture of the drone considered in this study.

**Figure 4 sensors-23-02291-f004:**
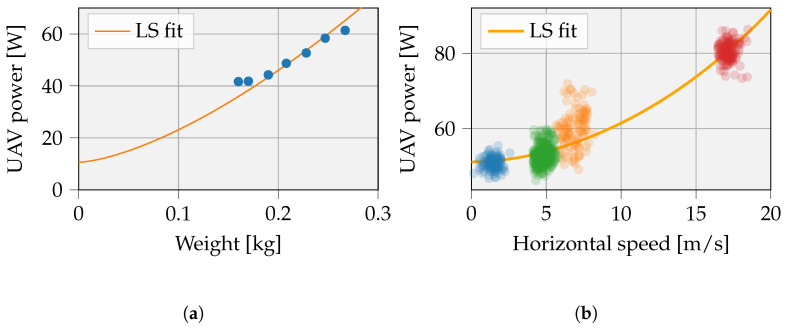
Experiment 1 gauged the hovering power related to UAV weight and payload. (**a**) LS fit on hovering data; Experiment 3 measured the UAV power and angle related to the horizontal speed. (**b**) LS fit on horizontal flight data. ($m_{\mathrm{payload}}$ = 50 g).

**Figure 5 sensors-23-02291-f005:**
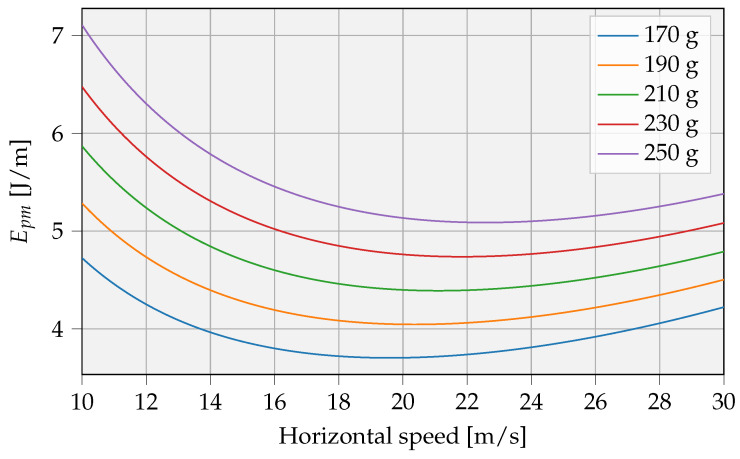
Energy per meter related to flight forward speed.

**Figure 6 sensors-23-02291-f006:**
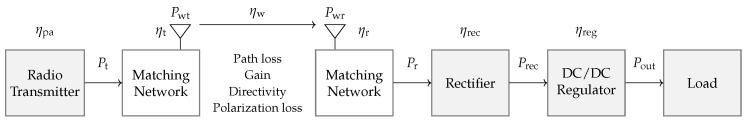
Transmission and efficiency model for RF power transfer [4].

**Figure 7 sensors-23-02291-f007:**
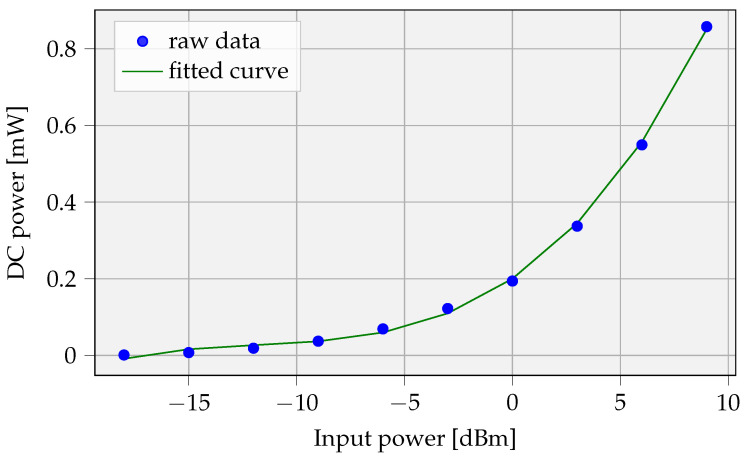
Fitted curve of the raw data of the AEM40940 energy harvester [22].

**Figure 8 sensors-23-02291-f008:**
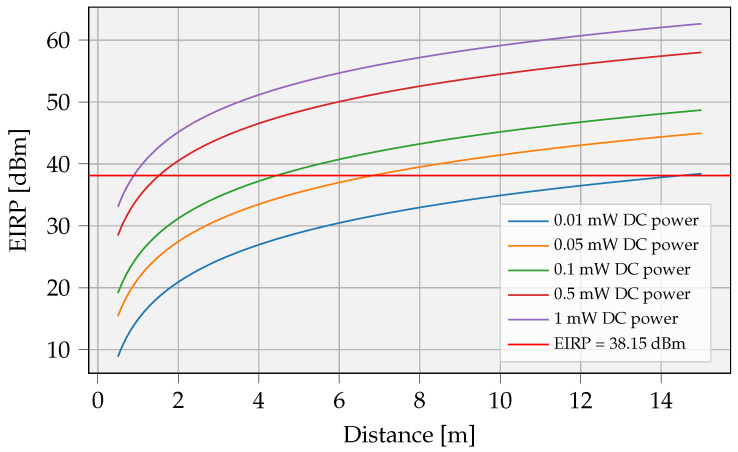
Required transmit power for an RF SISO system, expressed in EIRP.

**Figure 9 sensors-23-02291-f009:**
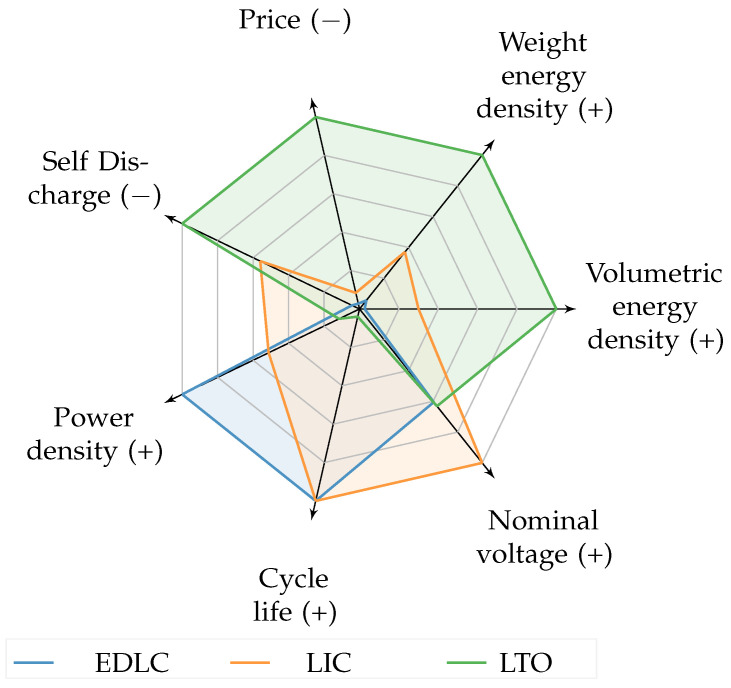
Spin chart comparing properties of energy storage compounds EDLC (orange) LIC (blue), and LTO (green). (+) higher is better, (−) lower is better.

**Figure 10 sensors-23-02291-f010:**
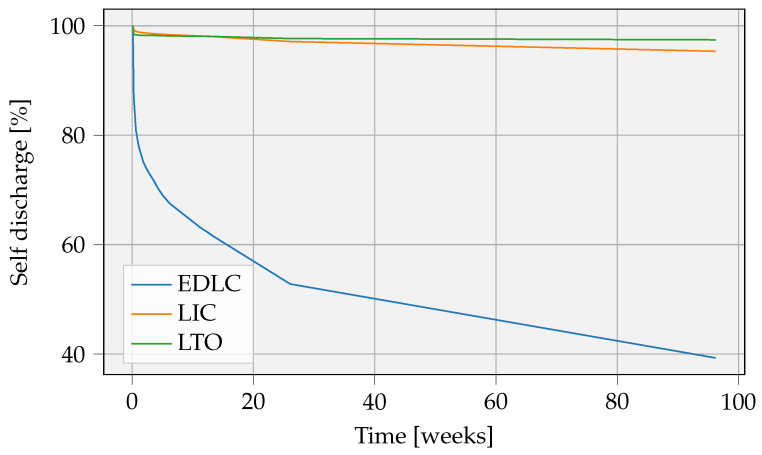
Self-discharge of EDLC, LIC, and LTO cells measured over approximately two years. The self-discharge is calculated as the percentage of remaining cell voltage compared with the initial voltage as a function of time.

**Figure 11 sensors-23-02291-f011:**
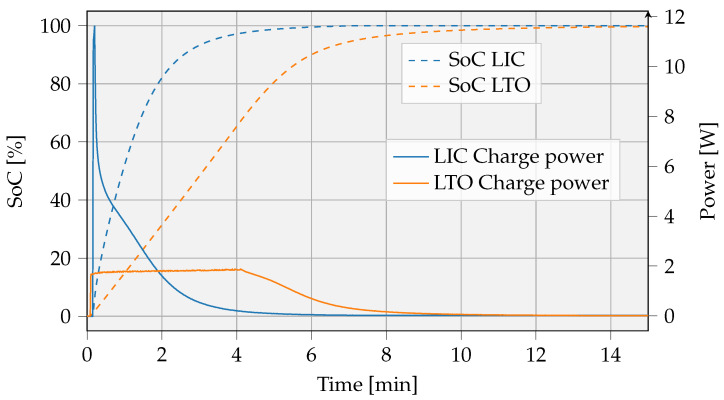
Measured charge curve of a 66.3 mAh LTO battery and a 50 F LIC capacitor.

**Figure 12 sensors-23-02291-f012:**
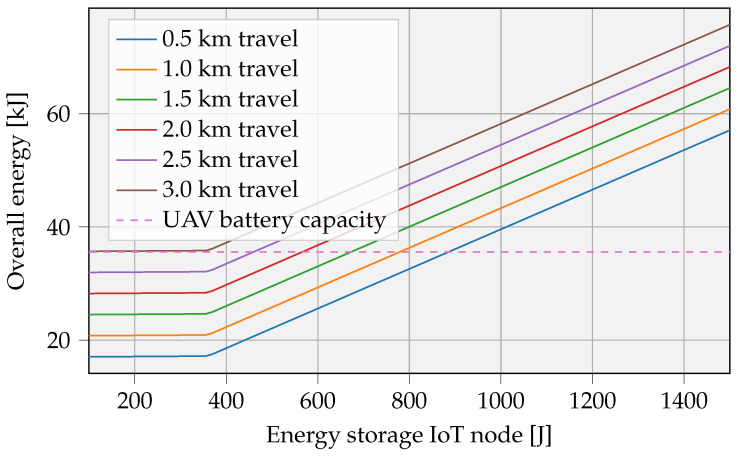
Maximum IoT storage capacity, limited by the battery capacity of the UAV, represented by a dotted line. The UAV battery capacity is 9.62 Wh, and the maximum charge power $P_{\mathrm{ch},\max}$ is set to 1 W. The energy storage IoT node involves an LTO cell with a charge rate of 10 C.

**Figure 13 sensors-23-02291-f013:**
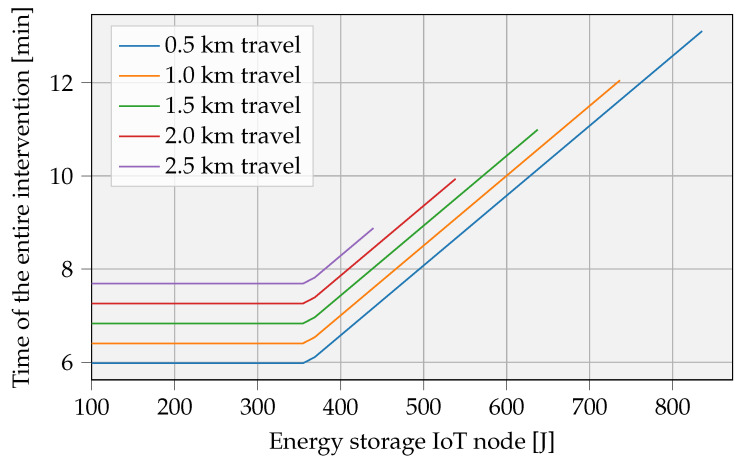
Overall operation time during one intervention. The same assumptions were adopted from Figure 12.

**Figure 14 sensors-23-02291-f014:**
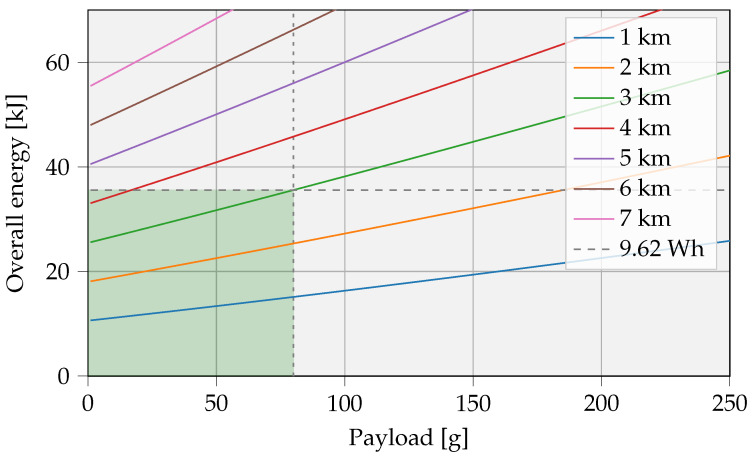
UAV energy needs for varying payloads and servicing distances. Delineated in green are the practically feasible cases, due to battery energy and payload limits. Assuming one UAV battery (9.62 Wh), the payload can be as high as 80 g.

**Figure 15 sensors-23-02291-f015:**
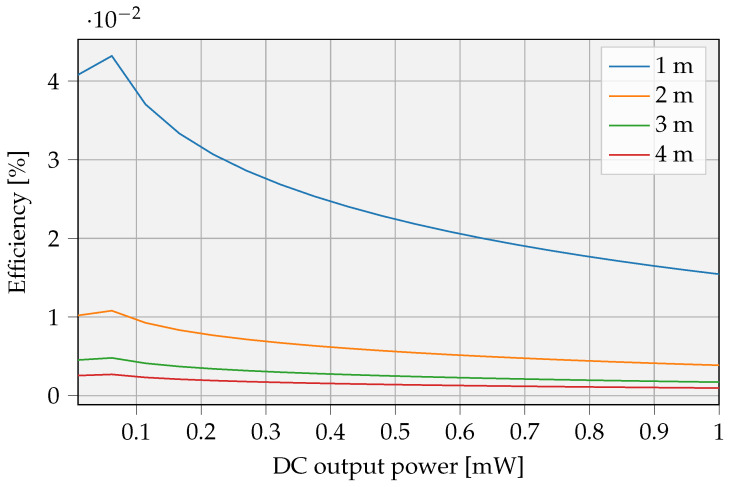
Estimate of the achievable efficiency levels for a SISO RFPT system with a pa with 77% efficiency. The maximum EIRP limit of 38.15 dbm is not taken into account.

**Figure 16 sensors-23-02291-f016:**
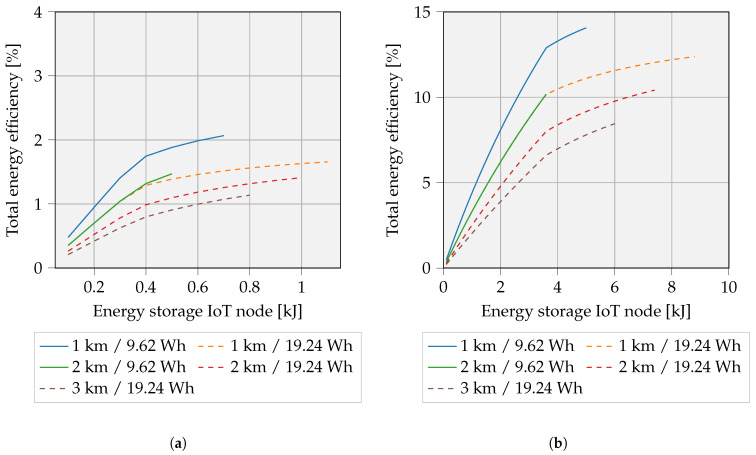
Overall energy efficiency including travel, WPT, and hover losses. A 3 km travel distance with a single UAV battery is not achievable, as also noted in previous analyses. (**a**) $P_{\mathrm{ch},\max}$ = 1 W; (**b**) $P_{\mathrm{ch},\max}$ = 10 W.

**Figure 17 sensors-23-02291-f017:**
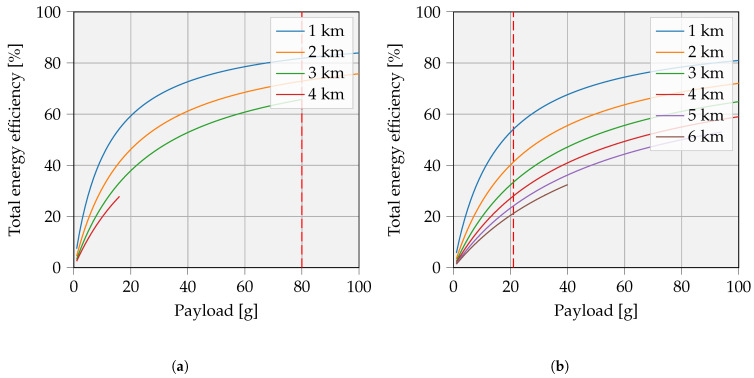
An NCA battery, with a weight energy density of 237 Wh/kg, assumed to be transported by the UAV. Assuming the total weight of the UAV should not exceed 250 g, the maximum payload of 80 g (one UAV battery) and 21 g (two UAV batteries) is indicated by the dotted vertical line. (**a**) One UAV battery; (**b**) two UAV batteries.

**Figure 18 sensors-23-02291-f018:**
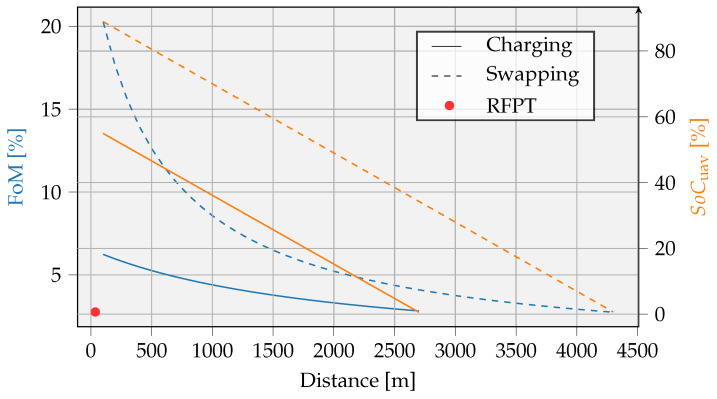
FoM comparison according to the parameters from Table 6.

**Figure 19 sensors-23-02291-f019:**
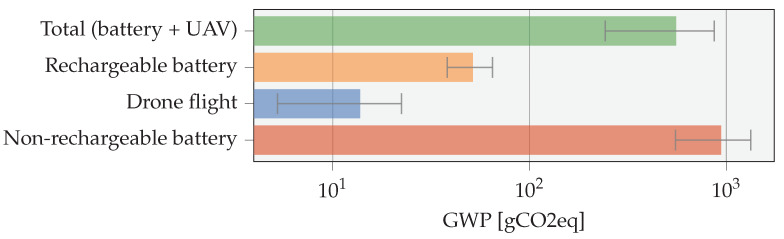
GWP of non-rechargeable fire-and-forget vs UAV-based servicing approach. Assumptions: 13 min flight at 50 Wh with lithium-based IoT node battery capacity of 2 kJ and an energy consumption of 20 J/day for 10 years. Note the logarithmic scale.

**Table 1 sensors-23-02291-t001:** Custom UAV parameters extracted from real-life experiments.

Parameter	Value [-]	Parameter	Value [-]
$k_{1}$	1.560	$c_{2}$	9.451
$k_{2}$	0.467	$c_{3}$	−0.0037
$c_{1}$	3.428	$c_{4}$	−0.0044

**Table 2 sensors-23-02291-t002:** Maximum harvesting distance 38.15 dBm EIRP related to varying DC output power levels.

DC Power (mW)	RF Input Power (dBm)	Max Distance (m)
(Required)	(Harvester)	(EIRP 38.15 dBm)
0.01	−14.15	14.5
0.05	−7.62	6.9
0.1	−3.88	4.7
0.5	5.43	1.6
1	10.06	0.9

**Table 3 sensors-23-02291-t003:** Calculated link efficiency using simulation data obtained via the finite element analysis for two vertical misalignments.

(x, y, z) [mm]	M [μH]	$L_{1}$ [μH]	$R_{1}$ [mΩ]	$L_{2}$ [μΩ]	$R_{2}$ [mΩ]	$k_{\mathrm{mrc}}\;$ [-]	$\eta_{\mathrm{mrc}}$ [%]
(0, 0, 100)	0.083	2.165	489	3.488	612	0.035	76.5
(0, 0, 150)	0.032	2.166	488	3.483	610	0.014	51.0

**Table 4 sensors-23-02291-t004:** Common rechargeable batteries with their weight energy density. The first four are lithium-based, while the latter two are nickel batteries.

Composition	Weight Energy Density [Wh/kg]	Reference
NMC	≈217	[33]
LFP	≈114	[34]
NCA	≈237	[35]
LTO	≈80	[31]
NiCd	≈36	[36]
NiMH	≈86	[37]

**Table 5 sensors-23-02291-t005:** Comparison of FoM values for different scenarios.

Cf. nr.	Scenario	Payload	IoT Battery Composition	# UAV Batteries	Distance [km]	Storage Capacity [kJ]	Remaining $SoC_{\mathrm{uav}}$ [%]	FoM Value [%]
1	Charging	0 g	LTO	1	2.5	1.8	0.3	5.1
	Swapping	80 g	NCA	1	2.5	68.3	14.4	69.2
2	Charging	59 g	LTO	2	3.7	5	0.8	7.1
	Swapping	64.9 g	NCA	2	6.3	5	7.8	7.1
3	Charging	59 g	LTO	2	1	4	46.5	10.5
	Swapping	63.7 g	NCA	2	1	4	80.0	22.0
4	Charging	59 g	LTO	2	4	4	6.5	6
	Swapping	63.7 g	NCA	2	4	4	39.4	8.5

**Table 6 sensors-23-02291-t006:** Proposed parameters to compare charging process against swapping process.

Parameters		Charging	Swapping
Size of depleted IoT battery	[J]	1000	1000
No. UAV batteries	(−)	1	1
$P_{\mathrm{ch},\max}$	[W]	10	N/A
$C_{\mathrm{rate}}$	(−)	10 (LTO)	N/A
$\eta_{\mathrm{wpt}}$	(%)	50	N/A
Weight energy density IoT battery	(Wh/kg)	N/A	237 (NCA)
$m_{\mathrm{payload}}$	(g)	0	1.17
$t_{\mathrm{hover}}$	(s)	324	60

**Table 8 sensors-23-02291-t008:** Comparison of UAV-based operation vs manual intervention [61,62].

	Car (Gasoline)	Car (Electric)	Walking	Cycling	Big UAV	Small UAV (Current Work)
Wh/km	562	187.5	65	17.5	22	1.39

## Data Availability

The data presented in this study are available on request from the corresponding author.

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
