# Peer review of "UAV-Based Servicing of IoT Nodes: Assessment of Ecological Impact"

_sensors, 2023, doi:10.3390/s23042291_

Round 1
Reviewer 1 Report
Please see the attached comments.

Reviewer 2 Report
See the file.

Reviewer 3 Report
The paper explores and studies three main techniques to increase the lifetime of the system. They compare the use of power transfer using fixed points in long-range links or using drones to transfer energy or using drones to change the batteries of the nodes.
For the case where the drone transfer energy to the node’s batteries, it can either land and then recharge or it can hover over the node to recharge in-flight,
but for practical reasons, only the in-flight recharging scheme is considered.
For the case where the drone changes the node’s battery, the authors also consider the case that it hovers over the node to swap the batteries.
A drone was created in order to obtain an energy consumption model for the flight operation of the UAV.
Additional to the theoretical energy consumption model found in the literature, the paper presents the results derived from three experiments using their own
drone in order to complete the existing models for their own application, i.e, energy transfer to nodes.
The paper is well-written and organized, it is very easy to follow. The mathematical analysis is sound and many variables are considered for the three presented scenarios. The results and equations derived in this work are of great importance for future research in this area.
I would suggest adding a figure comparing the performance of the three schemes to clearly see the one that is best suited for different scenarios like distance, number of nodes, packet reporting of the nodes (the more the nodes report packets, the more energy consumption in the system and the batteries have to be recharged more often), etc. The paper presents some tables comparing the Figure of Merit, but this is not very clear. A figure would be most representative.
Also, in eq. 20, the term on the left should be E_{total}, as stated in line 413.
Reviewer 4 Report
The paper compares the costs of IoT nodes under three different modes. The motivation is good and the paper is well organized. It is not an easy job to compare them from different aspects of a model. The results are informative and interesting. The topic is cutting-edge and can be interesting to some readers. I do not have major concerns as it presents in a good structure and I recommend a minor revision. Details are as follows.
The motivation of the paper is not clearly concluded in the introduction. "39 In this paper we study how unmanned vehicles (UVs) could help to address the above challenges,
40 focusing on new solutions to increase the lifespan of IoT devices." It can be better to replace "the above challenges" with a condensed description.
The second type of option "Engaging UVs deployed as mobile power bank, to charge batteries on-site." seems not related with Fig.1(2). How to understand the on-site charging. Do you mean wireless charging on a specific site? If so, you may want to emphasise WPT as a wireless type in the figure caption to avoid misunderstanding.
What is profile power in equ(1)? The author compares different models and mentions the proposed model is based on [19]. It is not easy to capture the highlights of the model without stating the novelty of the advancement of the proposed model. It can be mentioned above Fig.3 in short words.
The abbreviations such as UAV, IoT and others should be provided in full the first time it appears. The problems can be found in the abstract.
The authors mentioned abbreviations above the reference. But nothing is provided here, please check that.
Round 2
Reviewer 1 Report
The authors have basically addressed my concerns, no further comments.